# Immunotherapy for Multiple Myeloma

**DOI:** 10.3390/cancers11122009

**Published:** 2019-12-12

**Authors:** Hideto Tamura, Mariko Ishibashi, Mika Sunakawa, Koiti Inokuchi

**Affiliations:** 1Department of Hematology, Nippon Medical School, Tokyo 113-8603, Japaninokuchi@nms.ac.jp (K.I.); 2Department of Microbiology and Immunology, Nippon Medical School, Tokyo 113-8603, Japan; mariko-ishibashi@nms.ac.jp

**Keywords:** multiple myeloma, immunotherapy, antibody drug-conjugate (ADC), bispecific antigen-directed CD3 T-cell engager, chimeric antigen receptor T-cell (CAR-T) therapy, immune checkpoint inhibitor, tumor vaccine, allogeneic stem cell transplantation

## Abstract

Despite therapeutic advances over the past decades, multiple myeloma (MM) remains a largely incurable disease with poor prognosis in high-risk patients, and thus new treatment strategies are needed to achieve treatment breakthroughs. MM represents various forms of impaired immune surveillance characterized by not only disrupted antibody production but also immune dysfunction of T, natural killer cells, and dendritic cells, although immunotherapeutic interventions such as allogeneic stem-cell transplantation and dendritic cell-based tumor vaccines were reported to prolong survival in limited populations of MM patients. Recently, epoch-making immunotherapies, i.e., immunomodulatory drug-intensified monoclonal antibodies, such as daratumumab combined with lenalidomide and chimeric antigen receptor T-cell therapy targeting B-cell maturation antigen, have been developed, and was shown to improve prognosis even in advanced-stage MM patients. Clinical trials using other antibody-based treatments, such as antibody drug-conjugate and bispecific antigen-directed CD3 T-cell engager targeting, are ongoing. The manipulation of anergic T-cells by checkpoint inhibitors, including an anti-T-cell immunoglobulin and ITIM domains (TIGIT) antibody, also has the potential to prolong survival times. Those new treatments or their combination will improve prognosis and possibly point toward a cure for MM.

## 1. Introduction

Multiple myeloma (MM) is a hematologic malignancy characterized by the clonal proliferation of plasma cells that produce M-proteins, accompanied by various types of impaired immune function [1,2]. Despite treatments, including high-dose chemotherapy followed by autologous hematopoietic stem-cell transplantation (auto-SCT), and novel agents such as immunomodulatory drugs (IMiDs) and proteasome inhibitors (PIs), which have improved survival in MM patients over the past two decades, most patients become drug resistant and succumb to their disease [3,4]. IMiD (lenalidomide or pomalidomide)/PI (bortezomib or carfilzomib)-refractory patients who have received at least three prior lines of therapy regimens and have been exposed to an alkylating agent have a poor prognosis, with a median overall survival (OS) time of 13 months from the double-refractory state [5]. Further, almost all high-risk MM patients still have a poor prognosis, even after high-dose chemotherapy plus auto-SCT [6], and MM remains incurable even in standard-risk patients. Thus, strategies including immunotherapies are needed to maintain an enduring treatment response and to achieve a cure for MM patients.

Allogeneic stem-cell transplantation (allo-SCT) is regarded as one of the first immunotherapies offering the potential for prolonged survival time in MM patients [7,8]. Donor T cells may recognize minor histocompatibility antigen-presenting myeloma cells in the human leukocyte antigen (HLA)-matched setting, resulting in elimination of myeloma cells, the so-called graft-versus-myeloma effect [9]. Two studies showed longer progression-free survival (PFS) and OS times in MM patients treated with nonmyeloablative allo-SCT after auto-SCT compared with those treated with tandem auto-SCT [10,11], indicating that allo-SCT could be effective in a limited subset of patients with a small amount of residual myeloma cells after intensive chemotherapy, including auto-SCT. However, the treatment-related mortality rate was high at 15% [12]. Furthermore, there was no benefit to auto-SCT followed by nonmyeloablative allo-SCT in high-risk MM patients [13].

It was reported that expanded T-cell clones in MM patients were associated with longer survival and that most very long-term survivors without relapse had low levels of M-protein, suggesting that T-cell immunity plays a crucial role in inhibiting MM disease progression, similar to immunogenic tumors [14,15]. So far, various immunotherapies inducing myeloma-specific cytotoxic T-cell lymphocytes (CTLs), including immune checkpoint inhibitors and dendritic cell (DC)-based vaccines, have been reported to have some effect in a very limited subset of MM patients.

Recently, several antibody treatments have been shown to have specific effects in relapsed/refractory MM (RRMM) [16,17]. The anti-CD38 monoclonal antibody daratumumab (DARA), which can eliminate immunosuppressive cells and promote T-cell proliferation, was effective in inducing a deep response, especially in combination with IMiDs, for refractory RRMM patients [18,19].

Clinical trials using antibody drug-conjugate (ADC) and bispecific antigen-directed CD3 T-cell engager (BiTE) targeting myeloma-associated antigens are ongoing. To achieve more complete tumor elimination, gene-modified T-cell therapies, such as T-cell receptor (TCR)-engineered T-cell therapy and chimeric antigen receptor T-cell (CAR-T) therapy, were developed, and many trials are in progress. This review highlights the latest immunotherapies to treat MM.

## 2. Antibody-Based Immunotherapies

It is crucial to define desirable targets for the development of immunotherapy for MM patients. There are at least three definitions of the ideal myeloma-associated target antigen: 1) the expression of the target antigen is high at diagnosis and does not decrease with disease progression; 2) the target antigen should have a central function in the biology and/or pathophysiology of MM; and 3) the target antigen is expressed by as few normal tissues as possible. So far, the B-cell maturation antigen (BCMA) is the most reasonable immunotherapeutic target, and thus clinical trials of BCMA-targeting immunotherapies, including CAR-T, BiTE, and ADC, are ongoing with excellent results reported so far (Table 1) [20,21,22]. The results of antibody treatment targeting other immunotherapeutic antigens, i.e., CD19, CD38, CD56, CD138, SLAMF7, IL-6 receptor, VEGF receptor, immunoglobulin, integrin beta-7, and G protein-coupled receptor class C group 5 member D (GPRC5D), were reported [23].

### 2.1. IMiD-Intensified Antibody Treatment

Impaired immune surveillance, including increased numbers of immunosuppressive cells, is thought to be involved in MM disease progression at diagnosis and relapse. IMiDs have the dual effects of direct tumoricidal activity and immunomodulation, which improve immune function [24]. In high-risk smoldering MM patients, who have already exhibited impaired immune systems, lenalidomide/dexamethasone (Rd) was able to restore decreased expression of activation and type 1 T-helper proliferation markers and induced a marked shift in T cells and natural killer (NK) cells [25]. It was reported that the fold change in the number of regulatory T cells (Tregs) in peripheral blood was significantly higher in relapsed MM patients compared with patients who achieved a complete response (CR) 1 year after auto-SCT transplantation [26]. The expression level of CD38 on Tregs is higher than that on conventional T cells and in myeloma patients compared with healthy controls [27].

The anti-CD38 monoclonal antibodies have dual mechanisms: 1) effects through the ligation of tumor cells, i.e., antibody-dependent cell cytotoxicity (ADCC) mediated by NK cells, complement-dependent cytotoxicity (CDC), antibody-dependent cell phagocytosis (ADCP) mediated by macrophages, direct apoptotic induction by crosslinking, and inhibition of enzyme function; and 2) immune effects through inhibition of immunosuppressive cells, such as Tregs, regulatory B cells, and myeloid-derived suppressor cells (Figure 1) [1,28,29]. DARA monotherapy increased CD8^+^/CD4^+^ and CD8^+^/Treg ratios as well as T-cell clonality [28]. Lenalidomide can increase the CD38 expression level by promoting interferon-responsive sequence element transcription via Ikaros degradation [27] and enhance the antitumor effects of DARA-mediated ADCC and T-cell expansion, resulting in antimyeloma T-cell immunity with long-term antitumor effects. An investigation of the efficacy of DARA combined with Rd reported an overall response rate (ORR) of 93%, including a high CR rate of 43% in 569 RRMM patients with 1 median (range 1–11) prior regimen [18]. Furthermore, the percentage of responders who had long-term deep responses with sustained ≥12-month MRD negativity (10^–5^) was 13% in those who received DARA combined with Rd, but the percentage was smaller in those receiving other regimens, i.e., only 0.4% with Rd, 3% with DARA plus bortezomib/dexamethasone (Vd), and 0% with Vd [30]. That evidence of the immune effects of DARA was supported by a report by Kitadate et al., which showed that the absolute number of Tregs in peripheral blood was significantly higher in responders to DARA-based treatment compared with non-responders [31].

The antibody elotuzumab, a humanized IgG1 kappa immunostimulatory monoclonal antibody targeting SLAMF7 (also referred to as CS1 or CD319), in combination with lenalidomide or pomalidomide is used for the treatment of RRMM patients in clinical practice [32]. SLAMF7 is a member of the SLAMF family, which is expressed on the surface of various immune cells, such as T, B, and NK cells, but not in nonhematopoietic cells. It was reported that the expression levels of SLAMF7 mRNA were decreased in relapsed patients compared with newly diagnosed MM (NDMM) patients [33], and that, on the contrary, surface expression levels of SLAMF7 were almost the same at diagnosis and relapse [34]. SLAMF members share a similar structure with an extracellular domain consisting of Ig variable- and constant 2-like domains, and most members exhibit self-binding. The cytoplasmic domain contains an immunoreceptor tyrosine-based switch motif (ITSM), which can bind to adaptor proteins, resulting in modulation of immune responses. In NK cells, SLAMF7 self-ligation can enhance NK cell activity through adaptor protein EAT-2 recruitment [35], although the function and adaptor protein-mediated signaling of SLAMF7 in myeloma cells remain to be clarified. Recently, Kikuchi et al. have reported that the soluble form of SLAMF7 could increase myeloma cell proliferation in vitro and in vivo and that knockdown of surface SLAMF7 cancelled the proliferative function of soluble SLAMF7 in vitro [36]. They found that soluble SLAMF7 can bind surface SLAMF7 on myeloma cells, leading to activation of the SHP-2/ERK signaling pathways.

We also found that higher levels of serum soluble SLAMF7 were associated with advanced revised International Staging System (ISS) stage disease, and that the detection of serum soluble SLAMF7 was associated with a very low deep response rate. However, elotuzumab can rapidly neutralize soluble SLAMF7 in MM patients [37]. Xie’s group reported that SLAMF7 was upregulated in MM patients with the t(4;14) translocation because multiple myeloma SET protein (MMSET) regulates the SLAMF7 transcription level, and that SLAMF7 knockdown decreased myeloma cell colony formation [38]. Monotherapy with elotuzumab, which shows anti-MM efficacy via NK cell-mediated ADCC and augmented NK function, had no clinical effect in RRMM patients [39], which may be due to NK impairment in those patients. A phase III trial (*n* = 321) of elotuzumab combined with Rd demonstrated a very good partial response (VGPR) rate of 22% and ORR of 78% in RRMM patients and improved hazard ratios in the PFS and OS times of t(4;14)-positive RRMM patients compared with patients with the 17p deletion [40,41]. We reported the efficacy of elotuzumab combined with Rd in the real-world setting. The ORR was 56% and the clinical benefit rate was 79% for RRMM patients who had received a median of 3 prior therapies, which ranged from 1 to 12 [42]. Elotuzumab in combination with pomalidomide demonstrated a median PFS of 10.3 months with an ORR of 53% in RRMM patients who had received a median of 3 (range 2–8) prior therapies [43]. Those results demonstrated that SLAMF7 is associated with the pathophysiology of MM, and elotuzumab is an effective treatment in RRMM patients, especially those with the t(4;14) translocation.

### 2.2. Bispecific Antibodies

The results of clinical studies of the bispecific antigen-directed CD3 T-cell engager antibody BiTE appeared promising for the treatment of both bulky disease and MRD [44]. A CD19/CD3 bispecific antibody designed in the BiTE format, blinatumomab, was reported to be effective in patients with B-cell malignances, such as relapsed or refractory B-cell precursor acute lymphoblastic leukemia, chronic lymphocytic leukemia, and non-Hodgkin lymphoma [45,46,47]. The BCMA/CD3 bispecific T-cell engager induced myeloma lysis in vitro and in vivo [48,49]. Topp et al. showed that treatment with AMG 420, a BCMA bispecific T-cell engager in the BiTE antibody construct, induced MRD-negative CR in a phase I study of 42 RRMM patients who had received a median of 4 (range 2–13) prior treatment lines [21]. Cytokine release syndromes occurred in 38% (*n* = 16), with only one of grade 3. In the cohort treated with AMG 420 at the maximum tolerated dose of 400 µg/day (*n* = 10), the ORR was 70% and the MRD-negative sCR rate was 40%. We are awaiting the results of advanced clinical studies.

Compared with CAR-T therapy in vitro, BiTE-activated T cells showed similar functional avidity, as assessed by cytokine production of IL-2/TNFα and killing activity [50]. In the clinical setting, CD19/CD3-bispecific antibody has advantages, i.e., off-the-shelf administration with no preparation time needed. On the other hand, CAR-T therapy demonstrated higher response rates with deep responses in heavily pretreated patients [51]. A bispecific antibody, however, is less effective than anti-CD19 CAR-T therapy [52]. Other antigen/CD3 bispecific T-cell engagers are under development, and a clinical trial using a CD38/CD3 bispecific T-cell engager is ongoing [51]. New bispecific antibodies targeting FcRH5 and GPRC5D are being developed.

### 2.3. Immunochemotherapy and ADCs 

The safety, tolerability, and preliminary clinical activity of BCMA-ADC, a novel anti-BCMA antibody conjugated to the microtubule-disrupting agent monomethyl auristatin F (GSK2857916), were reported [22]. In RRMM patients (*n* = 35), including 20 (57%) heavily treated patients who had received ≥5 lines of therapy, the ORR was 60% (stringent CR 3%, CR 6%, VGPR 43%, and PR 9%) with a median PFS of 7.9 months. Grade 3 or 4 adverse events were reported in 28 (80%) of 35 patients, the most common of which were thrombocytopenia (4%) and anemia (14%). Other ADCs for MM patients are being examined in ongoing trials.

It was reported that most myeloma cells from NDMM and RRMM patients express high levels of SLAMF2 (CD48) [53,54]. Anti-CD48 monoclonal antibody can inhibit myeloma cell growth in vivo, suggesting it could be effective in treating MM patients. A phase I clinical trial (NCT03379584) using SGN-CD48A, a potent CD48-targeting ADC utilizing a novel glucuronide-monomethylauristatin E linker, is in progress, but patients are no longer being recruited. Similar to SLAMF2 and SLAMF7, SLAMF6 is highly expressed on myeloma cells from NDMM and RRMM patients [35,55]. In an MM xenograft model, SGN-CD352A, a humanized anti-CD352 engineered cysteine monoclonal antibody conjugated with 2 molecules of pyrrolobenzodiazepine dimer, a potent DNA-damaging cytotoxic drug, produced durable CRs. A safety study of SGN-CD352A treatment for RRMM patients is also underway (NCT02954796). SLAMF2/6-targeting ADCs are expected to show clinical benefits for RRMM patients. Phase 1 studies of other ADCs targeting CD38, CD46, and CD74 are ongoing.

## 3. Gene-Modified T-Cell Therapies

Gene-modified T-cell therapies consist of two categories: autologous T cells that express an affinity-enhanced TCR and a CAR recognizing known tumor target antigens [56]. A number of clinical trials using TCR-engineered T cells were performed in melanoma patients [57,58]. NY-ESO-1, an immunogenic cancer testis antigen, is expressed in up to 60% of advanced myelomas and its expression levels are higher in relapsed disease [56]. The testis antigen is associated with myeloma cell proliferation and high-risk features, i.e., cytogenetic abnormalities [59,60,61,62,63]. A clinical trial of NY-ESO-1/LAGE-1 TCR-engineered T cells showed clinical responses in 16 of 20 MM patients (80%) with advanced disease, with a median PFS of 19.1 months, although disease progression was observed with the loss of T-cell persistence or targeting antigen [64]. Further, TCR-engineered T-cell therapy has several disadvantages, including a requirement for a specific HLA type, leading to restrictions on treatment.

CAR-T therapy targeting CD19 (CART19, CTL019) has been established in heavily treated acute lymphocyte leukemia (ALL) patients because of high CR rates and durable effects in limited numbers of patients [65]. Anti-CD19 CAR-T therapy could replace allo-SCT in ALL, chronic lymphocytic leukemia, and B-cell lymphoma [66]. Several clinical trials of anti-CD19 CAR-T therapy in MM patients are in progress. BCMA-targeting CAR-T cell therapy (bb2121) demonstrated an excellent response, with an ORR of 85% and a median PFS of 12 months in heavily treated MM patients who had received a median of 7 (range 3–14) prior therapy regimens [20].

To enhance the effects of anti-BCMA CAR-T therapy, a CAR T-cell therapy targeting 2 BCMA domains (LCAR-B38M) was developed and demonstrated a high response rate, with an ORR of 88% (CR 68%, VGPR 5%, and PR 14%), MRD negativity of 63%, and median PFS of 15 months in 57 RRMM patents [67]. CD19-targeting CAR-T cell therapy after auto-SCT with lenalidomide maintenance showed responses of CR 10%, VGPR 60%, and PR 20%. CD19 expression is detected in very few myeloma cells by flow cytometry, although single molecule-sensitive direct stochastic optical reconstruction microscopy (dSTORM) demonstrated CD19 expression on a fraction of myeloma cells (10%–80%) in approximately 70% (*n* = 14) of MM patients with density of 13–5000 molecules/cell. CD19 CAR-T cells can eliminate those myeloma cells with a low density of <100 molecules/cell in vitro [68].

To increase the effect of BCMA CAR-T cells, CD19- and BCMA-targeting dual CAR-T cell therapy was developed and demonstrated a high response rates with a 95% ORR (stringent [s]CR 43%, CR 14%, VGPR 24%, and PR 14%) in 21 RRMM patients and an 100% ORR (sCR 70%, CR 10%, and VGPR 20%) and 60% MRD negativity (<10^−6^) in 10 high-risk NDMM patients [69,70]. CD19- and BCMA-targeting dual CAR-T cell therapy is thought to be a reasonable treatment to eliminate a wide range of heterogenetic myeloma cells in patients. BCMA is known to derive from myeloma cells in active disease, and thus BCMA-targeting CAR-T combined with a γ-secretase inhibitor prolonged survival in a murine myeloma model because the inhibitor upregulated BCMA expression levels on MM cells [71,72]. A phase I clinical trial (NCT03502577) of this combination in relapsed or persistent MM patients is now in progress. In the CARTITUDE-1 phase 1b/2 clinical trial of BCMA CAR-T using an identical construct of LCAR-B38M, the ORR was 100% (≥CR 69%, ≥VGPR 86%) and the MRD-negative rate was 82% with two severe CRS’ (one of grade 3 and one of grade 5) [73]. Other CAR-T clinical trials targeting BCMA are ongoing, i.e., dual-specificity CD38 and BCMA CAR-T (NCT03767751), PD1-CAR-T-secreting mutant PD-1Fc fusion protein with high-affinity binding to the PD-L1 ligand (NCT04162119), autologous CD8+ T-cells expressing anti-BCMA (NCT03448978), etc.

The effects of anti-BCMA CAR-T therapy (bb2121) were investigated in high-risk MM patients who had adverse chromosomal abnormalities, high tumor burden, and extramedullary disease [20]. In terms of adverse events, cytokine release syndrome (CRS) is important in the care of patients treated with CAR-T therapy. Clinical trials of CTL019 anti-CD19 CAR-T therapy showed a severe CRS occurrence rate of 29% in B-ALL, 39% in B-CLL, and 20% in B-non-Hodgkin’s lymphoma patients, although BCMA-targeting CAR-T included a grade 3/4 CRS occurrence rate of 6% (all grades 76%) in a bb212 trial and 7% (all grades 90%) in a LCAR-B38M trial. Similar to CRS, the frequency of neurotoxicity was lower: bb2121 42% (grade 4: 3% = reversible), LCAR-B38M 2% (grade 1: aphasia, agitation, and seizure-like activity).

At present, the clinical data on CAR-T therapies targeting CD138, NKG2D ligands, and kappa-light chain remain insufficient [74,75,76]. Clinical trials using other antigen-targeting CAR-T cell regimens in MM, which are targeting CD38, SLAMF7, CD44v, CD56, GPRC5D, transmembrane activator and CAML interactor (TACI), Lewis Y, and NY-ESO-1, as well as preclinical studies targeting CD229 (SLAMF3), integrin β7, CD70, and CD1d, are in progress [23,34,77,78,79]. CAR-T therapy may find a place in MM treatment in the near future.

## 4. Immune Checkpoint Inhibitors

Exhausted T cells expressing immune checkpoint receptors are involved in disease progression in cancer models, although the role of exhausted T cells in the pathophysiology of MM remains controversial [15,80,81]. An in vivo murine myeloma model demonstrated that the expression levels of immune checkpoint receptors such as programmed cell death protein 1 (PD-1), T-cell immunoglobulin and ITIM domains (TIGIT), lymphocyte activation gene-3 (LAG-3), and T-cell immunoglobulin mucin-3 (Tim-3) were increased on CD8^+^ T cells in bone marrow in MM-relapsed mice compared with those in control mice and MM-controlled mice [82]. PD-L1 (CD274) was first identified as a homologue of the B7 family and named B7-H1 by our group [83,84]. PD-L1, which is expressed on most tumor cells, can inhibit antitumor T-cell responses [83]. The PD-1–PD-L1 pathway is involved in tumor growth with evasion from tumor immunity.

We and others previously reported that PD-L1 expression levels are higher on plasma cells from MM patients compared with those in healthy controls and monoclonal gammopathy of undetermined significance (MGUS) patients and its expression is often upregulated in the relapsed/refractory phase [85,86]. Furthermore, high PD-L1 expression levels on MM cells are associated with disease progression in asymptomatic MM patients and MM cells with MRD [87,88]. PD-L1 expression on MM cells is upregulated by the IL-6 signal cascade through STAT3, MEK1/2, and JAK2, and by IFN-γ produced by CTLs and NK cells through the MEK/ERK pathway [85,86], suggesting that PD-L1 expression can be induced in the myeloma microenvironment. We also found that PD-L1^+^ myeloma cells had more proliferative potential and were resistant to antimyeloma agents, with higher expression levels of Ki-67 and Bcl-2 compared with PD-L1^−^ myeloma cells [86]. Furthermore, PD-1-bound PD-L1 can activate the Akt pathway in myeloma cells, leading to drug resistance [89]. Those data suggest that PD-L1 plays a crucial role in the pathophysiology of MM. In addition, PD-1 expression on T cells was reported to be correlated with tumor burden in a murine myeloma model [90,91,92]. In MM patients, the percentage of CD8^+^PD-1^+^ T cells co-expressing CD57 was significantly increased by approximately 35% in comparison with healthy controls (about 20%) [81]. In relapsed MM patients one year after auto-SCT, PD-1 expression levels on both CD28-CD4^+^ and CD28-CD8^+^ T cells were significantly increased compared with those in patients sustaining CR [26].

In contrast to the expected efficacy, the anti-PD-1 immune checkpoint inhibitor nivolumab alone had few effects in 63% of 27 RRMM patients with stable disease, which was maintained for a median of 11.4 weeks [93]. This might have been because the immune system in RRMM is more exhausted, including impaired T-cell tumor immunity. Only one RRMM patient who also underwent radiation therapy (RT) during the treatment period achieved CR after nivolumab retreatment for two months [93]. RT can synergize with immune checkpoint inhibitors because cell death induced by RT is thought to induce DC activation through “eat-me (calreticulin-CD91)”, “danger (HMGB1-TRL4)”, and “find-me (ATP-P2RX7)” signals, resulting in enhanced CTL function (Figure 2) [94]. The combination of RT and anti-PD-1/PD-L1 monoclonal antibody may be effective in both RRMM and lung cancer patients [95]. The IMiD lenalidomide can enhance antimyeloma immune responses, even when induced by immune checkpoint inhibitors, suggesting the clinical benefits of this combination [96,97]. The anti-PD-1 antibody pembrolizumab in combination with the IMiD lenalidomide or pomalidomide demonstrated a high overall response rates of 50%–60% in heavily treated patients with a median 3 to 4 prior chemotherapies [98,99,100]. A phase II clinical trial of pembrolizumab combined with pomalidomide/dexamethasome showed an sCR/CR rate of 8%, VGPR rate of 19%, and PR rate of 33% in 48 RRMM patients. However, overall survival was decreased in two randomized trials of pembrolizumab in both NDMM and RRMM patients. Krauss provided an overview of the US Food and Drug Administration analysis, demonstrating that there was no difference in the ORR with or without immune-related adverse events (irAE) in RRMM patients, whereas an increased ORR was observed in patients with irAE with an increased irAE rate in NDMM patients [101]. Further studies are needed to clarify the optimal use of immune checkpoint inhibitors in RRMM patients.

Recently, research into other immune checkpoints, including TIGIT, LAG3, and TIM3, has focused on the immunosuppressed status of MM patients. Exhausted effector T cells expressing CD28^–^LAG3^+^TIGIT^+^ were increased in MM patient’s refractory to DARA combined with pomalidomide [102]. The percentages of TIGIT^+^CD8^+^ T cells and their expression levels were increased in patients with NDMM and RRMM [82]. In an in vivo murine myeloma model, CD8^+^ T cells in the bone marrow of MM-relapsed mice expressed higher levels of TIGIT and lower levels of another receptor of CD155, DNAM-1, which can deliver a positive signal in T cells through the binding of CD155, compared with those in control or MM-controlled mice. Anti-TIGIT monoclonal antibody decreased myeloma cells in bone marrow and prolonged survival compared with control-Ig or anti-PD-1 monoclonal antibody-treated mice [103], suggesting that TIGIT expression is more dominant than PD-1 in the immunosuppressive function in MM. A clinical trial using anti-TIGIT monoclonal antibody in advanced solid tumors is underway (NCT04047862, NCT03563716). A new checkpoint inhibitor, an anti-TIGIT antibody, is a potential efficient treatment for MM patients.

## 5. Immune Vaccination: Dendritic Cell Vaccination and Tumor-Antigen Peptide Vaccination

Tumor vaccines against tumor-associated antigens were reported to be promising in eliminating MRD without significant toxicity [104,105]. However, there are very few reports of tumor vaccines for MM patients. Idiotype-pulsed DC vaccination after auto-SCT prolonged OS but not PFS in 27 MM patients compared with historical controls [106]. The administration of a DC/myeloma fusion vaccine, which is a very attractive technology because DCs can express many patient tumor antigens and elicit efficient CTL activity to patients who had undergone auto-SCT, resulted in the achievement of CR/near CR in 24% of the PR group after transplantation [107]. Immune checkpoint inhibitors such as anti-PD-1 in combination with a fusion vaccine is appears to be a desirable immunotherapy for MM patients with stable disease and MRD [108]. A clinical trial (NCT01067287) of anti-PD-1 antibody plus the DC/myeloma fusion vaccine following SCT is still ongoing.

## 6. Conclusions

Profound disrupted immune surveillance is caused by dysfunctions of NK cells, CTLs, and DCs with increased immunosuppressive cell levels in MM patients. Improved understanding and elucidation of myeloma biology and pathophysiology have led to landmark achievements in drug development. IMiDs, key drugs in treating MM, can enhance the effect of monoclonal antibody treatments, such as DARA. Other antibody-based immunotherapies, such as bispecific antibody, CAR-T and ADCs, show excellent efficacy in RRMM patients, even heavily treated ones, but outcomes are not yet satisfactory owing to the loss of target antigens and T-cell exhaustion. To achieve the desired treatment breakthroughs, many immunotherapies, including CAR-T regimens, are under development for MM patients. The hope is that those treatments will prolong OS in patients with no or low levels of MRD. It was reported that immunotherapies such as DC vaccine and allo-SCT can achieve longer survival with small doses of M-protein in some patients. Thus, it is also necessary to develop immunotherapies that can inhibit MM cell proliferation, even in the relapse phase after CAR-T and bispecific antibody therapies. The results so far suggest that T-cell immunity is crucial in controlling the disease, and thus immunotherapies are required to treat MM patients.

## Figures and Tables

**Figure 1 cancers-11-02009-f001:**
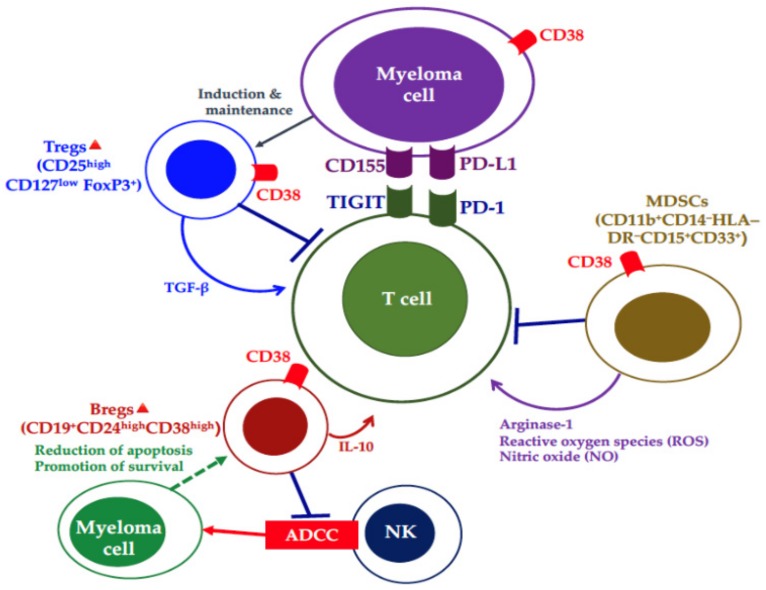
Immune evasion and anti-CD38 monoclonal antibody-targeting cells in the myeloma microenvironment. T-cell immune function is inhibited by the surrounding immunosuppressive cells, such as regulatory T cells (Tregs), regulatory B cells (Bregs), and myeloid-derived suppressor cells (MDSCs), signaling from immune checkpoint receptors. Anti-CD38 antibodies can eliminate myeloma cells as well as immunosuppressive cells.

**Figure 2 cancers-11-02009-f002:**
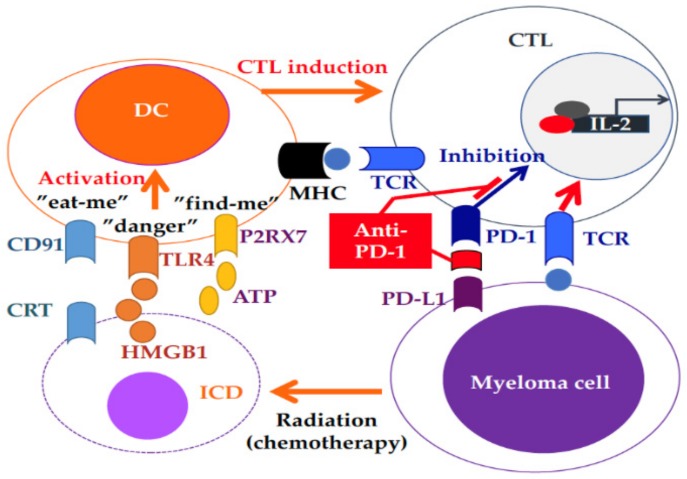
Combined immunotherapy. Radiotherapy (RT) can induce immunogenic cell death, resulting in dendritic cell (DC) activation through “eat-me (calreticulin-CD91)”, “danger (HMGB1-TRL4)”, and “find-me (ATP-P2RX7)” signals, leading to enhanced cytotoxic T-cell lymphocyte (CTL) function.

**Table 1 cancers-11-02009-t001:** B-cell maturation antigen (BCMA)-targeting immunotherapies for refractory myeloma patients.

	CAR-T	Bispecific antibodies	ADC
Off-the-shelf	Not yet	Yes	Yes
Ease of administration	+	+ ~ ++	++++
Dependent on patient T cell condition	Yes	Yes	No
Results of representative clinical trials			
**Protocol**	**Bb2121** **(*n* = 33)**	AMG420 (n = 42)	GSK2857916 (n = 35)
Median age (y) (range)	58 (37–74)	63	**60 (40–75)**
**Prior treatment lines**	**Median 7** **(range 3–14)**	**Median 4** **(range 2–13)**	**≥5 prior lines** **57%**
Response	ORR 85% MRD(–) CR 45%	ORR 70% MRD(–) CR 40%	ORR 60%
Median PFS	**12 months**	**9 months**	8 months
Major toxicity	Neutropenia 85%, anemia 45%, thrombocytopenia 45%, CRS 76% (grade 3: 6%), neurologic toxic effects 42%	CRS: all grades 38% (severe CRS 2%), serious peripheral neuropathy 5%	Grade 3–4 AEs 80%; corneal AEs (vision blurred, keratitis, photophobia, dry eye, keratopathy, eye pain), thrombocytopenia

ORR, overall response rate; CR, complete response; MRD, minimal residual disease; PFS, progression-free survival; CRS, cytokine release syndrome; AEs, adverse events.

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
