# Peer review of "Immunotherapy for Multiple Myeloma"

_cancers, 2019, doi:10.3390/cancers11122009_

Round 1
Reviewer 1 Report
Nice, timely paper.
My remarks:
Page 1: HDM and auSCT is no 'new' treatment anymore. I would remove the word new.
Page 1: was it not patients IMID and PI refractory and treated with an alkylater have a described OS of 13 months in leukemia 2017, Kumar te al?
Page 2: The anti-CD38 monoclonal antibody daratumumab (DARA), which can eliminate immunosuppressive cells and promote T-cell proliferation, in combination with IMiD was shown to be effective in inducing to gain a long-term complete response (CR) rates with no detection of minimal residual disease (MRD) [18,19].
Please adapt sentence.
page 2: also gprc5d is a target, already in clinical trial.
Page 2, table 1. Why only these ones mentioned?
Page 3: SALMF7. Please improve.
page 6: CAR-T therapy for MM patients may replace allo-SCT in the near future. This is not a correct statement. For ALL allo SCT is performed in the relapsed setting, but for MM, with all the alternative regimens, this is not an advised treatment modality anymore, outside of clinical trials, see international guidelines. SO it will not replace. Although it might well be of course that CAR T will have a place in MM treatment in the nearby future.
Why discussing just a subset a BCMA CAR T's for MM. Please add the results for the other BCMA CAR T's already used in phase 2 and 3 clinical studies.
Also Daratumumab is mentioned for immunomodulation, however this is also the case for other CD38 mAbs. Please elaborate on this.
Can you discuss about all of the patients relapsing on CAR T and bispecifica therapies in MM, while this is not the case for DLBCL and ALL were we do see a plateau in survival curves. The manuscripts states that these kind of therapies might lead to cure, but that is not the case (yet), as you can clearly see in the survival curves. Can you please elaborate on this. And also maybe not suggest the way to cure (yet)
Author Response
We thank the reviewers for their comments on our manuscript and appreciate their help in improving it. We have revised the manuscript in response to each comment. The revised text appears in red. We hope that the revisions are satisfactory and enable the manuscript to be accepted for publication in Cancers.
1) Page 1: HDM and auSCT is no 'new' treatment anymore. I would remove the word new.
Thank you for your suggestion. We removed the word "new" (page 1, line 29).
2) Page 1: was it not patients IMID and PI refractory and treated with an alkylaterhave a described OS of 13 months in leukemia 2017, Kumar et al?
We corrected the sentence pointed out (page 1, lines 32–36).
3) Page 2: The anti-CD38 monoclonal antibody daratumumab (DARA), which can eliminate immunosuppressive cells and promote T-cell proliferation, in combination with IMiD was shown to be effective in inducing to gain a long-term complete response (CR) rates with no detection of minimal residual disease (MRD) [18,19]. Please adapt sentence.
The sentence was revised (page 2, lines 57–60).
4) page 2: also gprc5d is a target, already in clinical trial.
It was noted in the CAR-T section that G protein-coupled receptor class C group 5 member D (GPRC5D) is a target of CAR-T therapies for myeloma patients. We added a GPRC5C paper[#77. Smith EL,et al Sci Transl Med. 2019:27;11(485)]in the References.Additionally, Kodama T et al. reported the results of anti-GPRC5D/CD3 bispecific T-cell-redirecting antibody in vitro and in vivo [Mol Cancer Ther 2019;8(9):1555-1564], and thus we mentioned that GPRC5D is also a target in the introduction of the "2. Antibody-based immunotherapies" section (page 2, line 76). Other new targets are also added as follows: Bispecific Abs: New bispecific antibodies targeting FcRH5, GPRC5D, are developing (page 5, lines 173–174).ADCs: Phase 1 studies of other ADCs targeting CD38, CD46, and CD74 are ongoing (page 5, line 194).
5) Page 2, table 1. Why only these ones mentioned?
We tried to compare the effects and safety of immunotherapy approaches. Many clinical trials of CAR-T are ongoing, although most of them have only reported immature data so far. Therefore we mentioned the representative data on each immunotherapy. To understand the difference, we added other general points such as off-the-shelf, ease of administration, and dependent on patient T cell conditionto compare those therapies.
6) Page 3: SALMF7. Please improve.
We edited SLAMF3 (page 3, line 128).
7) page 6: CAR-T therapy for MM patients may replace allo-SCT in the near future. This is not a correct statement. For ALL allo SCT is performed in the relapsed setting, but for MM, with all the alternative regimens, this is not an advised treatment modality anymore, outside of clinical trials, see international guidelines. SO it will not replace. Although it might well be of course that CAR T will have a place in MM treatment in the nearby future.
According to your comments, we edited the sentence "Similar to B-cell malignances, CAR-T therapy for MM patients may replace allo-SCT in the near future" to "CAR-T therapy may find a place in MM treatment in the near future"(page 6, lines 253–254).
8) Why discussing just a subset a BCMA CAR T's for MM. Please add the results for the other BCMA CAR T's already used in phase 2 and 3 clinical studies.
In this manuscript, we described the results of bb2121 (phase 1), LCAR-B38M (phase 1), and CD19- and BCMA-targeting dual CAR-T (Phase 2). We could not find the results of phase 2 and 3 clinical studies using other BCMA CAR-T with one exception. In the CARTITUDE-1 phase 1b/2 clinical trial of BCMA CAR-T using an identical construct of LCAR-B38M, the ORR was 100% (≧CR 69%, ≧VGPR 86%) and the MRD-negative rate was 82% with 2 severe CRS (one of grade 3 and one of grade 5) [Madduri, D.et al].We also added information from ongoing clinical trials using BCMA, i.e., dual-specificity CD38 andBCMA CAR-T(NCT03767751), PD1-CAR-T-secreting mutant PD-1Fc fusion proteinthat shows high-affinity binding to the PD-L1 ligand (NCT04162119), autologous CD8+ T-cells expressing an anti-BCMA (NCT03448978), etc. (page 6, lines 233–239).
9) Daratumumab is mentioned for immunomodulation, however this is also the case for other CD38 mAbs. Please elaborate on this.
As pointed out, other CD38 mAbs have also immunomodulatory effects. Thus, we changed daratumumab to anti-CD38 monoclonal antibodies in that sentence (page 3, line 94).
10) Can you discuss about all of the patients relapsing on CAR T and bispecifica therapies in MM, while this is not the case for DLBCL and ALL were we do see a plateau in survival curves. The manuscripts states that these kind of therapies might lead to cure, but that is not the case (yet), as you can clearly see in the survival curves. Can you please elaborate on this. And also maybe not suggest the way to cure (yet)
As you mentioned, it is important to induce a plateau in the survival of MM patients. Actually, some long survivors after allo-SCT have sustained small amounts of malignant plasma cells with no proliferation and symptoms. Thus, we added the sentences including this in the Discussion (page 8, lines 342–346).
Reviewer 2 Report
I have reviewed the manuscript entitled “Immunotherapy for multiple myeloma”. In this manuscript, you reviewed several kinds of novel immunotherapies including chimeric antigen receptor T – cell (CAR-T) therapy, immune checkpoint inhibitors. The manuscript is well written, and the concept proposed by the authors is very interesting.
I have only minor comments.
There is not described on an official appellation of “TIGIT” in abstract, line 20. In addition to that, there are not described on official appellations of PD-1, TIGIT, LAG-3, and TIM-3, in line 251 as well. These abbreviations should be stipulated in the manuscript.I think the content of “exhausted effector T cells” from line 110 to line 111 overlaps the content of the end of “4. Immune checkpoint inhibitors”, from line 296 to line 298. You should remove the former sentence (from line 110 to line 111).
Author Response
We thank the reviewers for their comments on our manuscript and appreciate their help in improving it. We have revised the manuscript in response to each comment. The revised text appears in red. We hope that the revisions are satisfactory and enable the manuscript to be accepted for publication in Cancers.
1) There is not described on an official appellation of “TIGIT” in abstract, line 20. In addition to that, there are not described on official appellations of PD-1, TIGIT, LAG-3, and TIM-3, in line 251 as well. These abbreviations should be stipulated in the manuscript.
We added the definitions of TIGIT, PD-1, LAG-3, and TIM-3 in the Introduction and text (page 1, lines 20-21; page 6, lines 259–261).
2) I think the content of “exhausted effector T cells” from line 110 to line 111 overlaps the content of the end of “4. Immune checkpoint inhibitors”, from line 296 to line 298. You should remove the former sentence (from line 110 to line 111).
According to your suggestion, the former sentence was removed.